# Measuring of the Energy Expenditure during Balance Training Using Wearable Electronics

Tomáš Veselý *, Markéta Janatová, Pavel Smrčka, Martin Vítězník, Radim Kliment and Karel Hána

Department of Information and Communication Technologies in Medicine, Faculty of Biomedical Engineering, Czech Technical University in Prague, 128 00 Prague, Czech Republic; marketa.janatova@fbmi.cvut.cz (M.J.); smrcka@fbmi.cvut.cz (P.S.); martin.viteznik@fbmi.cvut.cz (M.V.); radim.kliment@fbmi.cvut.cz (R.K.); hana@fbmi.cvut.cz (K.H.)
* Correspondence: tomas.vesely@fbmi.cvut.cz; Tel.: +4-202-2496-8574

**Abstract:** Homebalance Stability medical device, based on audio-visual feedback and Nintendo Wii Balance Board, is a suitable tool for telerehabilitation of balance issues in patients with brain damage. The main goal was to expand the system by energy expenditure measurements and to verify the usability of the telemetric mobile device FlexiGuard. We used the FlexiGuard system (developed at our institute) and Oxycon (JAEGER® Oxycon Mobile, Germany) to measure the energy expenditure. We performed measurements on eight probands. Each proband underwent six activities for a total length of 90 min. During these activities, we measured energy expenditure using Oxycon and heart rate using the FlexiGuard system, from which we calculated the energy expenditure. By comparing the energy expenditure from measuring the heart rate with the FlexiGuard system with that from the Oxycon reference device, we verified the applicability of the FlexiGuard system for estimation energy expenditure. The average deviation from the reference instrument was under 30%. The conventional method, such as Oxycon, cannot be used during home therapy. Therefore, we upgraded the platform of our telemetry system (FlexiGuard), which can measure the heart rate and calculate the energy expenditure.

**Keywords:** telerehabilitation; force platform; audio-visual feedback; energy expenditure





## 1. Introduction

The use of audio-visual feedback and game-like training in the neurorehabilitation of patients with brain damage has made remarkable progress in the past decade. The use of devices based on gaming technology increases the effectiveness of therapy and patient motivation [1,2] and proves a helpful supplement to conventional therapies. The new technologies include motion sensors, robotic systems, virtual reality, physiological function sensors, and telemedicine applications. At present, the necessary neurorehabilitation is primarily provided in rehabilitation centers. Our vision is to make it accessible in patients' homes because home therapy based on audio-visual feedback, without direct involvement of a therapist, can save difficult-to-move patients the trip to the outpatient clinic, reduce financial costs, save personnel, make home therapy more attractive, and increase the frequency and quality of therapeutic units. Individual home therapy provided utilizing telemedicine solutions could target much larger groups of indicated patients without compromising the quality of care. The resulting large amount of data transmitted from the patients' homes to the medical center where they are integrated, enlarged, processed, and analyzed can provide a precious source of relevant information [3]. This information can help to set up an optimum neurorehabilitation process.

Balance disorders in neurological patients worsen self-sufficiency when performing routine daily activities and increase the risk of falling. The force platform and audio-visual feedback [4] are often used in the therapy of balance disorders using technical means. Methods based on virtual reality technology and feedback are a valuable supplement to conventional neurorehabilitation approaches. Many studies have investigated the effect of

motion sensor therapy and audio-visual feedback, which has been assessed as suitable for use in neurological patients [5–10].

For this purpose, we have developed therapeutic systems in our laboratory based on audio-visual feedback and the Nintendo Wii balance board (see Figures 1 and 2) [11]. The force platform, originally designed for the gaming industry, is also used in rehabilitation in outpatient and home settings [5–7,10,12]. The Nintendo Wii balance board has useful parameters for application in the therapy and diagnostics of balance disorders [13]. A patient standing on the force platform controls the gaming scene through the changes in his center of gravity position. The center of pressure (COP) movement is visualized in a virtual environment to provide visual feedback. The difficulty of training and sensitivity of the sensors can be adjusted with respect to the individual patient's condition. The patient's condition is objectively evaluated initially and throughout the rehabilitation process. The effectiveness of the patient's COP is measured and presented as a score value. The data is analyzed in time and frequency areas and the figures are saved. The audio-visual feedback is represented by 3D realistic virtual scenes (crossing the road, finding one's way inside a building . . . ) or through simple 2D game tasks. In the laboratory environment, this therapy has proven impact on changes in the heart rate, skin resistance, and other parameters in patients with balance disorders due to brain damage [3].

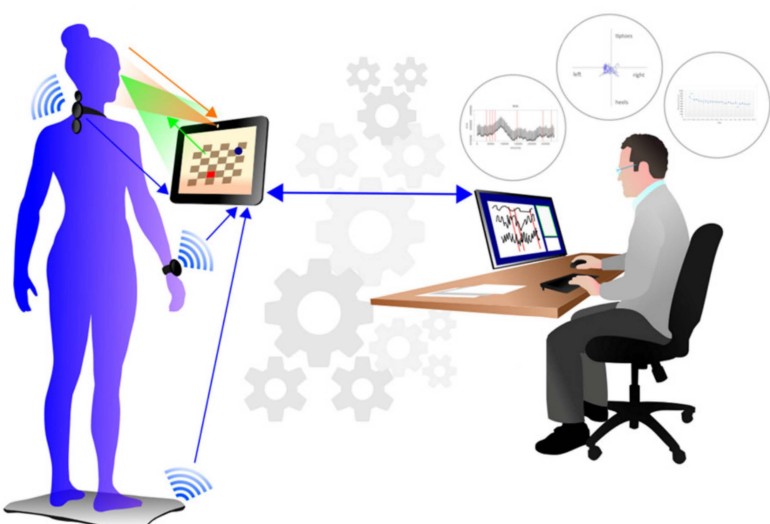

**Figure 1.** Scheme of the use of wearables when using medical device Homebalance.

The patient's regular individual exercise in the home environment is crucial for the optimum neurorehabilitation process. In order to increase efficiency and safety, we are developing a telerehabilitation solution, which is designed both for the therapy of balance disorders and for the monitoring of a patient's condition during this therapy. The main component of the telerehabilitation system that we have been developing is the Homebalance Stability medical device, which we created for home therapy of balance disorders. Pilot studies have demonstrated that the system can be used in the home therapy of patients without adverse effects and with positive impacts on their dynamic postural stability [11,14].

The system we are developing can be further innovated and supplemented with additional sensors. One of the goals is to monitor patients' energy expenditure (EE) during rehabilitation. Current conventional methods measuring energy expenditure (such as direct and indirect calorimetry or the double-labelled water method) cannot be used during rehabilitation, especially in-home care. These methods require special and expensive equipment such as a calorimetric chamber, breathing gas analyzer, or laboratory analyses of the urine sample. The use of such complex methods is therefore impossible for the intended

purpose. Instead, however, it is possible and very easy to use wearable electronics, also called "wearables" [15,16].

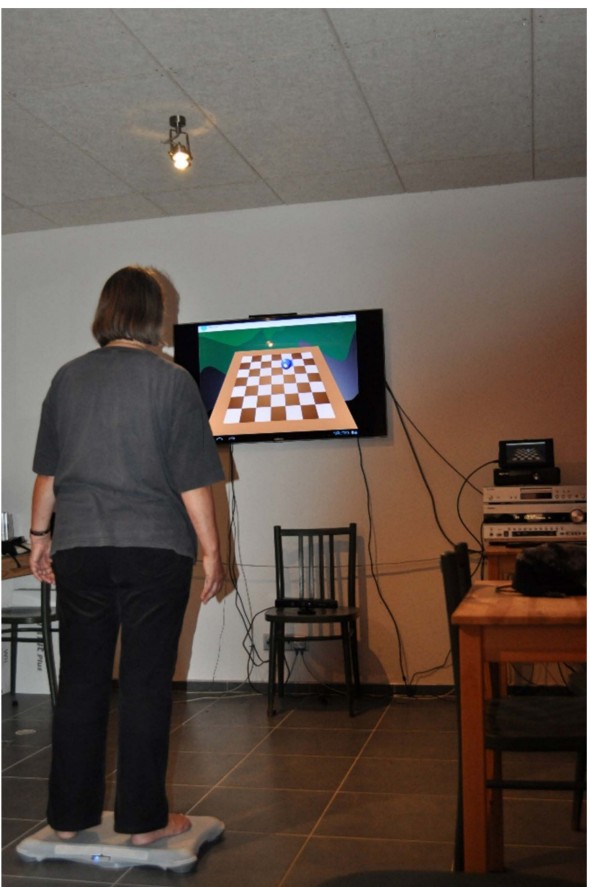

**Figure 2.** Patient during therapy with Homebalance.

Patients with balance disorders are at a higher risk of falling and decompensating their health condition than the healthy population. Neurological patients may experience epileptic seizures when audio-visual feedback is used if under excessive physical strain. Therefore, it is crucial to keep energy expenditure at an optimum level during exercise at home. Overload can easily occur if there is no therapist to supervise the exercise.

An individual's energy expenditure is a significant indicator. It can be used, for example, in the training of Integrated Rescue System (IRS) units and soldiers, in sport practices, and also in addressing current population diseases such as obesity.

The study's main goal is to expand the system by energy expenditure measurements and verify the usability of the telemetric mobile device FlexiGuard. FlexiGuard is a wireless monitoring system for measuring biosignals developed at our institute. The FlexiGuard system consists of a chest belt (see Figure 3) that scans heart rate (including individual RR intervals for further heart rate variability analysis), motion activity, the temperature on the skin, temperature and humidity under clothes, and GPS location. These signals are stored on a memory card inside the device with a frequency of 5 Hz for data analysis and simultaneously are transmitted by a wireless network for online preview. The unit is controlled by an ARM Cortex-M3 microcontroller. Up to 30 devices can be monitored by a single visualization unit. Data are broadcasted via an 868 MHz mid-long range wireless module. The typical wireless communication range is kilometers in open areas and hundreds of meters in urban areas like cities. Several receivers can receive data simultaneously, which further increases the range of wireless communication. The device is powered by a Li-Pol battery with a capacity of 2000 mAh, and the battery life per charge is over 24 h. The device may further be equipped with additional nodes that can measure,

for example, temperature or physical activity elsewhere in the body. Communication with these nodes takes place using a 2.4 GHz WBAN network based on ANT technology. These nodes were not used during this study. More detailed information about the system is given elsewhere [17]. Using the FlexiGuard device is entirely safe. It only requires a chest belt that does not affect the user, similar to conventional commercial sports testers, and therefore there is no danger to the patient.

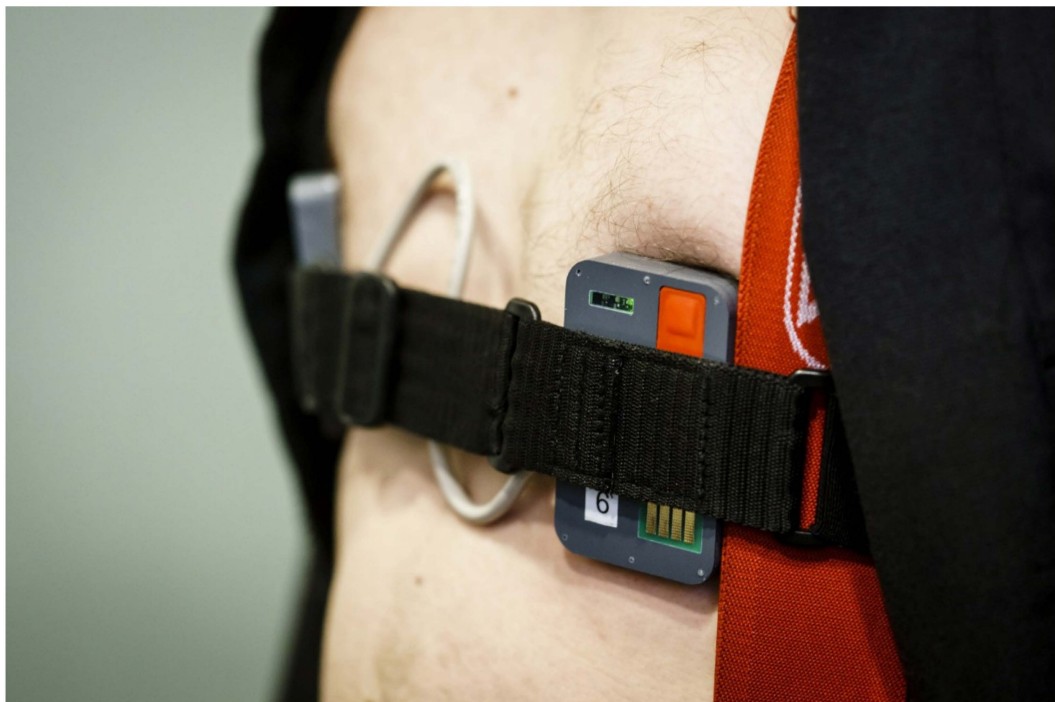

**Figure 3.** Chest belt of the FlexiGuard monitoring unit used during the study.

## 2. Materials and Methods

To verify whether the FlexiGuard system can be used to determine the energy expenditure, we measured healthy probands. Written informed consent was obtained from all participants, and the study the local Ethics Committee approved the study. All procedures performed in studies were in accordance with the ethical standards of the institutional research committee and with the 1975 Helsinki declaration and its later amendments. Each proband went through about 90 min of measurement. During the measurement, they performed 6 steady activities (bed rest, flat walking, climbing stairs, arm work, light load on an ergometer—Ergo 1, higher load on an ergometer—Ergo 2), during which we measured energy expenditure using Oxycon (JAEGER® Oxycon Mobile, Hoechberg, Germany, system measuring undirect calorimetry using breathing mask and gas analyzer) and also heart rate using FlexiGuard (Department of Information and Communication Technologies in Medicine, Faculty of Biomedical Engineering, Czech Technical University in Prague, Prague, Czech Republic). Subsequently, we calculated the energy expenditure. We then compared the results of both measurements, Oxycon being the reference method against which we benchmark the energy expenditure figures from the heart rate measured by FlexiGuard. The comparison of the calculated figures of energy expenditure as opposed to the measured expenditure for one proband is shown in Figure 4.

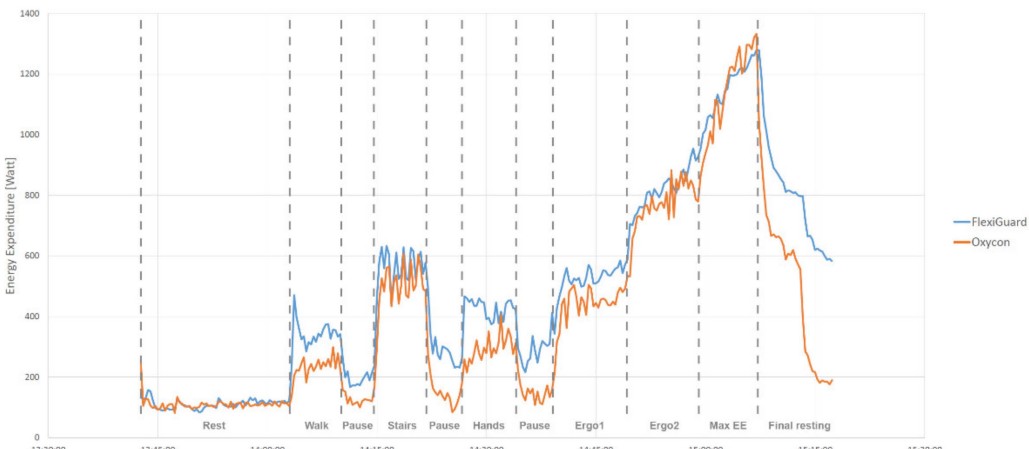

**Figure 4.** Benchmark of energy expenditure graph for one proband (blue curve—FlexiGuard, orange curve—Oxycon).

The FlexiGuard system scans a single-channel electrocardiogram (ECG) from the chest belt using a 24-bit analogue front-end, with a sampling frequency of 1000 Hz. The heart rate is calculated from the ECG using a QRS detector (based on the Pan–Tomkins algorithm [18], which is considered one of the gold standards in QRS detection). The data is stored on an SD card and sent through a wireless interface to a PC and is thus available online during therapy. The energy expenditure is calculated from the heart rate.

The energy expenditure is calculated using a series of relationships. We calculate the resting energy expenditure $EE_{rest}$ [Watt] using the Milfin equation [19]:

$$EE_{rest} = \frac{[(9.99 \cdot m) + (6.25 \cdot h) - (4.92 \cdot a) + 5] \cdot 4.19}{1440} \tag{1}$$

for men, respectively:

$$EE_{rest} = \frac{[(9.99 \cdot m) + (6.25 \cdot h) - (4.92 \cdot a) - 161] \cdot 4.19}{1440} \tag{2}$$

for women, where $m$ is mass [kg], $h$ is height [cm], $a$ is age [years].

Using the energy equivalent of oxygen, we can calculate the resting oxygen consumption $VO2_{rest}$:

$$VO2_{rest} = \frac{EE_{rest}}{20.1} \cdot 1000 \tag{3}$$

Maximum oxygen consumption $VO2_{max}$ is calculated based on Uth et al. [20]:

$$VO2_{max} = \frac{HR_{max}}{HR_{rest}} \cdot 15 \cdot m, \tag{4}$$

where $HR_{max}$ is maximum of heart rate [beats/min], $HR_{rest}$ is the heart rate at rest [beats/min], and $m$ is mass [kg].

From heart rate, using the Equations (2), (3) and the Astrand–Ryhming nomogram [21], we can calculate current oxygen consumption $VO2$:

$$VO2 = \left[ \frac{HR - HR_{rest}}{HR_{max} - HR_{rest}} \cdot (VO2_{max} - VO2_{rest}) \right] + VO2_{rest}, \tag{5}$$

where $HR$ is the current heart rate [beats/min].

Then the energy expenditure $EE$ from the energy equivalent of oxygen is calculated:

$$EE = \frac{VO2 \cdot 20.1}{60} \tag{6}$$

The estimate of energy expenditure can be determined with different accuracy depending on the level of individualization of the input data. The least accurate is to use average data (mass, height, age) for the entire population. $HR_{max}$, $HR_{rest}$ is derived from these general data. The estimate will be refined if the basic data (mass, height, age) about the person being measured will be known. This information will give more accurate values for $HR_{max}$, $HR_{rest}$, and $EE_{rest}$. The calculation is then refined if $HR_{max}$ and $HR_{rest}$ to the person are accurately measured. The most accurate result is achieved if measurements are performed on FlexiGuard and Oxycon for individual activities and a calibration curve is derived from the results. The latter option is not suitable for rehabilitating patients with brain damage, as the use of Oxycon could be risky for them. For this reason, we have chosen to measure healthy people when we know their basic information and $HR_{rest}$ is obtained when the patient is at rest in bed, and $HR_{max}$ when driving on an ergometer at a higher load. However, we are aware that it is not possible to obtain $HR_{max}$ in this way in patients, but it is necessary to use one of the generally known equations.

The probands underwent six stages: low EE (rest), medium EE (walking, arms, Ergo 1), and high EE (stairs, Ergo 2). During the resting stage, the probands were instructed to be as calm as possible, as if trying to fall asleep lying on their backs. The measured persons walked along a straight corridor 30 m long without elevation during the walking stage. Another station (arms) was focused on lifting weights in the range of 2–6 kg depending on the physical fitness of individuals. Subsequently, the measurement took place while walking up and down the stairs. In the end, the probands underwent an ergometer ride, first at low load (Ergo 1, men 80 Watts, women 60 Watts), then at higher loads (Ergo 2, men 160 Watts, women 120 Watts). Each proband was always to take a 5-min rest between individual stages while sitting on the chair. The measurement time intervals were as follows: rest—20 min, walking—7 min, rest on the chair—5 min, stairs—7 min, rest on the chair—5 min, arms—7 min, rest on the chair—5 min, Ergo 1—10 min, Ergo 2—10 min, then every 3 min increase the load on the ergometer by 30 watts for women and by 50 for men until exhaustion, rest on the chair—10 min.

## 3. Results

Eight (6 men, 2 women) probands were included in the study. Their average age was $27.6 \pm 3.9$ years and average body mass index (BMI) was $24.9 \pm 3.7$. All participants went through 6 stages. Each stage was divided into 3 intervals, so we obtained 24 values for each measured stage of energy expenditure. The results are presented as relative errors between the calculated values with FlexiGuard and the measured values with Oxycon (see Table 1).

**Table 1.** Results of individual stages.

| Stage | Relative Error (%) | | | One Sample $t$-Test $p$-Value | |
|---|---|---|---|---|---|
| | **Median** | **Min** | **Max** | **Mean = 0** | **Mean ≥ 0** |
| Rest | 32.63 | −4.71 | 103.01 | <0.001 | 1 |
| Walking | 34.95 | −14.26 | 78.17 | <0.001 | 1 |
| Stairs | 8.59 | −17.97 | 27.43 | 0.071 | 0.96 |
| Hands | 63.26 | 24.59 | 148.57 | <0.001 | 1 |
| Ergo 1 | 25.33 | 1.68 | 54.08 | <0.001 | 1 |
| Ergo 2 | 15.03 | −3.24 | 45.59 | <0.001 | 1 |

The stages with higher energy expenditure had less variance in the relative error than those with low EE. The lowest EE values were achieved within the resting stage. On the contrary, the highest EE values were reached during the Ergometer 2 or the stairs stage.

During the measurement, the FlexiGuard measuring system had no adverse effects. Probands rated the use of the FlexiGuard chest strap as without difficulties and the level of comfort as good or very good. On the other hand, the use of the Oxycon system was considered annoying or very annoying, proving the assumption that its use in therapy is not possible for patients with motor disorders.

During the measurement of 8 probands for 90 min each, there was not a single failure of the heart rate and EE measurement in FlexiGuard. We evaluated the statistics in the R software (version 3.6.0 by The R foundation, Auckland, New Zealand) using a one-sample *t*-test, where we examined the hypothesis that the relative deviation of the measured values from the actual value is not statistically significantly different from zero. We confirmed the normality of the data using the Shapiro–Wilk test.

The *t*-test results showed that the hypothesis that the mean value of the relative deviation of the measured data is zero could not be rejected only for the stairs stage. The results from the other five stages confirmed that the mean value of the relative deviation is different from zero. This is visible from Figure 4. Based on this, we tested once again using the *t*-test whether the deviation of the measured data is "statistically significantly higher than zero", as the relative deviation may be negative in our data. Moreover, the result for all stages, including Stairs, is YES, see Table 1. It follows that our EE measurement method overestimates the results, see Figure 5.

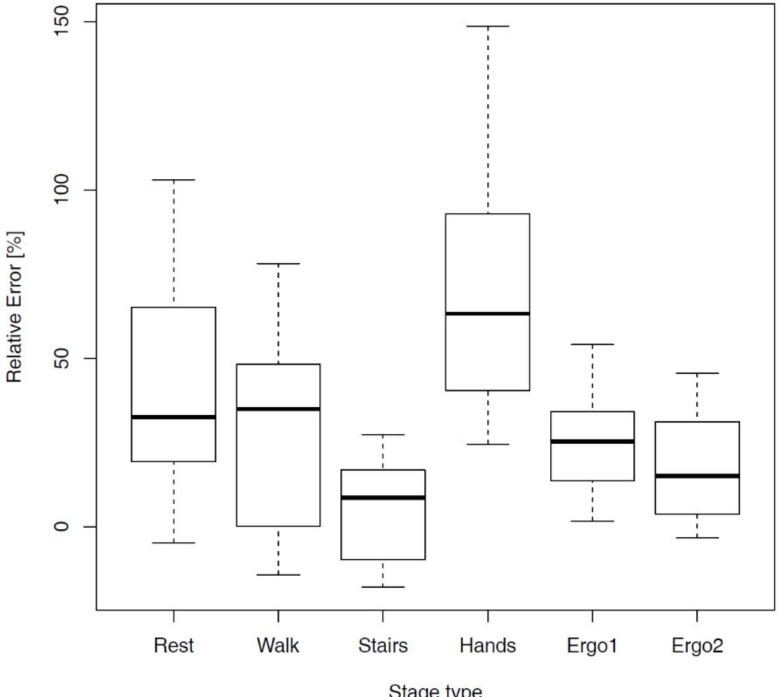

**Figure 5.** Graph showing calculated relative errors for each stage.

## 4. Discussion

There are several methods by which energy expenditure can be determined very accurately. These include direct and indirect calorimetry or the double-labelled water method. However, these methods are hardly applicable during field and semi-field measurements, during day-to-day activities, as well as during rehabilitation on a force platform (direct calorimetry requires placing an individual into a measuring room—calorimetric chamber; indirect calorimetry requires a face-fitted breathing mask).

With the advancement of modern wireless WSN and WBAN networks, it is possible to estimate energy expenditure using some of the sensors operating within these networks, like a heart rate sensor or an accelerometer—a motion sensor, to name just a few. These systems can be used in normal day-to-day activities, but also during practices, training, rescue team operations, or rehabilitation of patients. They limit the user much less than the conventional methods described above.

Heart rate measurement systems can estimate an individual's energy expenditure. These systems are benchmarked against reference methods, suggesting that it is still neces-

sary to keep refining these systems and possibly looking for new sensors, methods, and procedures suitable for determining an individual's energy load.

To enable the use of the telerehabilitation system that we are developing, in everyday clinical practice, it is important to enable automatic energy expenditure telemonitoring utilizing devices and real-time data evaluation during exercise. For this reason, it was not possible to use Oxycon Mobile (JAEGER® Oxycon Mobile, Hoechberg, Germany), which we used in laboratories to determine energy expenditure. Instead, we used the FlexiGuard system [17,22] also developed at our institute.

We compared the FlexiGuard energy expenditure measurement system with the reference indirect calorimetry method using Oxycon (JAEGER® Oxycon Mobile, Hoechberg, Germany). Benchmark measurements could not be performed on patients because it is dangerous. Patients must be able to move all their limbs without restriction and breathe freely. Failure to observe optimal conditions may deteriorate their health condition and cause other problems, such as epileptic seizures. For this reason, we validated the FlexiGuard system by applying it to a healthy population.

In a meta-analysis of the studies, Peng et al. [23] found that the energy expenditure using the Wii Balance Board is comparable to normal physical activities at moderate or medium load in a healthy population [23]. Exercise is more demanding for patients with motor deficiency than for a healthy population. Therapy using the board was assessed as more demanding than conventional therapy in a study aimed at comparing the effects of therapy. Patients need to focus more on therapy and must take rest breaks during therapy [24]. Given the above, it would not be appropriate to have a healthy population practice on the Wii Balance Board when validating FlexiGuard. The strenuousness of this exercise for a healthy population is very low compared to that for patients with motor function disorders. We have therefore devised an alternative, a more than 90-min-long test comprising low, medium, and high load activities, during which the energy expenditure values measured by FlexiGuard were benchmarked against those obtained through the indirect calorimetry reference method.

Our system has determined the energy expenditure from the heart rate. By comparing it with the Oxycon reference device (JAEGER® Oxycon Mobile, Hoechberg, Germany), we verified the applicability of the FlexiGuard system for determining the energy expenditure. The deviation from the reference device was, on average, under 30% for all six stages and 23% excluding hands load. One possible explanation is that the formula was created based on a population different from the one existing today. Over the past 60 years, the population is likely to have become "lazier". Therefore, the variables used in the formula are no longer valid for today's population and need to be reviewed. In addition, the formulas are based on data of young athletes more than 60 years ago. The graphs show that the greatest deviation relates to the arm work. One explanation for this is that all other stages mainly involve the work of the legs. Legs have a large muscle volume, and therefore, maximum performance is limited by the heart's potential. When doing leg work, everyone stops upon reaching their maximum heart rate, after which they can no longer boost performance because the heart will not supply more oxygen to the muscles. Arm work is different as arms do not have nearly as much muscle volume and, therefore, the performance is limited by the maximum potential of the arm muscles. Despite the fact that during the arm work stage, the exertion was very high and near the maximum limit of what the probands were able to handle for 7 min, their heart rate and EE measured by Oxycon were at about the level of normal walking. The heart thus still had some potential, but the arms could no longer cope. Due to the exertion and possibly also pain in the arms, the heart rate increased due to stimuli other than just the work of the arms, and thus at this stage, we could probably see the highest overestimation of the measured data.

The probands underwent six stages. The stages with higher energy expenditure had less variance in the relative error than those with low EE. This is due to the fact that at low pulse rates, the pulse rate is influenced by factors other than energy expenditure alone. Additionally, obtaining maximum and minimum heart rate values is probably not entirely

accurate. Between individual stages, each proband was always to take a 5-min rest while sitting on the chair, which was inserted in order to prevent the individual stages from affecting each other. Clearly, after a physically demanding stage such as stairs, a 5-min break may not be sufficient in some cases. However, the aim of the experiment was not to accurately measure the energy expenditure of each stage but to determine whether FlexiGuard can be used to estimate energy expenditure.

## 5. Conclusions

The use of conventional methods such as Oxycon (JAEGER® Oxycon Mobile, Hoechberg, Germany) is not suitable for home-based telerehabilitation. Therefore, the platform was upgraded by our telemetry system (FlexiGuard), which can measure pulse frequency and accurately calculate energy expenditure.

We verified the applicability of the telemetry system FlexiGuard system for measuring pulse frequency and accurately calculating energy expenditure. The usability during measurements was verified on eight probands for over 90 min each. No single adverse event or effect was detected during the measurement. The measuring part of the FlexiGuard system is merely a chest strap that patients can easily wear, as they are hardly affected by it. FlexiGuard is a feasible method for measuring energy expenditure during telerehabilitation.

## 6. Patents

Hána, K.; Kašpar, J.; Kučera, L.; Mužík, J.; Smrčka, P.; Veselý, T.; Vítězník, M.

Supervision device for monitoring people, especially in difficult conditions and a sensor allocation system on the human body.

Czech Republic. Patent CZ 307930. 17 July 2019.

Available from: https://isdv.upv.cz/doc/FullFiles/Patents/FullDocuments/307/307930.pdf (accessed on 17 July 2019).

**Author Contributions:** Conceptualization, T.V. and M.J.; methodology, T.V., P.S. and K.H.; software, T.V., M.V. and R.K.; validation, T.V., M.J. and P.S.; formal analysis, T.V., M.V.; investigation, R.K., M.J. and T.V.; resources, M.J.; data curation, T.V.; writing—original draft preparation, T.V.; writing—review and editing, M.J. and K.H.; visualization, T.V.; supervision, K.H.; project administration, P.S.; funding acquisition, T.V., P.S. and K.H. All authors have read and agreed to the published version of the manuscript.

**Funding:** This research was funded by the Grant Agency of the Czech Technical University in Prague, grant no. SGS17/207/OHK4/3T/17 (Research and development of new methods and models of energy expenditure estimation using wearables) and SGS17/206/OHK4/3T/17 (Complex monitoring of the patient during the virtual reality-based therapy).

**Institutional Review Board Statement:** The study was conducted in accordance with the Declaration of Helsinki, and approved by the Institutional Review Board (or Ethics Committee) of Faculty of Biomedical Engineering, Czech Technical University in Prague (protocol C5/0117 date of approval 15 May 2017).

**Informed Consent Statement:** Informed consent was obtained from all subjects involved in the study.

**Data Availability Statement:** Not applicable.

**Acknowledgments:** The authors would like to thank all the probands who underwent the measurement.

**Conflicts of Interest:** The funders had no role in the design of the study; in the collection, analyses, or interpretation of data; in the writing of the manuscript, or in the decision to publish the results.

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
