# Peer review of "Measuring of the Energy Expenditure during Balance Training Using Wearable Electronics"

_electronics, doi:10.3390/electronics11071096_

Round 1

Reviewer 1 Report

Broad comments. The authors have made a concise overview of the topic and a brief reference to existing literature. They have indicated the main task of the paper among its motivation. Finally, they have pointed out the key message and the potential benefits of their work.

Specific comments. In general, the text is very well structured and has clearly defined topics. Some comments for improvement:

  1. As a general drawback, I could say that there is no reference to similar wearables (e.g. [1]) in different health monitoring areas where that could also serve an alternative (e.g. vessels).

[1] Daskalos A-C, Theodoropoulos P, Spandonidis C, Vordos N. Wearable Device for Observation of Physical Activity with the Purpose of Patient Monitoring Due to COVID-19. Signals. 2022; 3(1):11-28. https://doi.org/10.3390/signals3010002

  1. More or less all fundamental theory details that are needed are discussed and a review of the problem under evaluation is sufficient. It would be beneficial to clarify the reasoning and the behind the selection of the system used. Would it be dangerous to wear something near heart if relevant problems exist?
  2. Besides, authors could refine the introductory section such that the use of Nintendo Wii based equipment in the current work is clearer. For example, a large part of the work is related with the description of this equipment but is not used?
  3. Authors could better describe the technical details of their used wearable solution (FlexiGuard), even though two references are provided with relevant details.
  4. Do authors think that the sample of volunteers is statistically important?
  5. While in the discussion section authors describe the reasoning for deviations presented in table 1, did authors try to use different (updated) coefficients for the equations?

Reviewer 2 Report

As the authors mentioned, the main goal of this study was to expand the system by energy expenditure measurements and to verify the usability of their innovated telemetric mobile device (FlexiGuard) for patients who have balance disorders and need rehabilitation (therapy) at home. The authors compared the energy expenditure graph between their system with a commercial system (Oxycon) to strengthen that verification. In the reviewer’s opinion, the provided data and information in this study are not sufficient.  However, this study was well written and there are some sections which can be improved. Therefore, the reviewer suggests that this study after improvement can be acceptable for publication in Electronics journal.

Here are some comments for further improvement of the manuscript:

  • Looking at Figures 1-3, why were the positions of some sensors different between the scheme (on neck + wrist) and the photos (on chest)?
  • In Figure 4, please indicate the stages in the graph (for example: resting, walking, resting, etc. or I, II ,I, etc.)
  • In Figure 4, the baseline drift of FlexiGuard is quite significant and it started after the walking stage and kept increasing more and more. The authors did not explain clearly about that in line 260-269. However, that is not clear why the baseline drifted like that. Certainly, the heart rate of the volunteers increases gradually without sufficient relaxation. Therefore, the reviewer suggests the authors to perform an additional test with longer resting time or change the orders of some stages.
